# Pan-Cancer Analysis of the Genomic Alterations and Mutations of the Matrisome

**DOI:** 10.3390/cancers12082046

**Published:** 2020-07-24

**Authors:** Valerio Izzi, Martin N. Davis, Alexandra Naba

**Affiliations:** 1Faculty of Biochemistry and Molecular Medicine, University of Oulu, FI-90014 Oulu, Finland; Valerio.Izzi@oulu.fi; 2Finnish Cancer Institute, 00130 Helsinki, Finland; 3Department of Physiology and Biophysics, University of Illinois at Chicago, Chicago, IL 60612, USA; mdavis48@uic.edu; 4University of Illinois Cancer Center, Chicago, IL 60612, USA

**Keywords:** extracellular matrix, tumor microenvironment, copy number alterations, mutations, protein domains, survival

## Abstract

The extracellular matrix (ECM) is a master regulator of all cellular functions and a major component of the tumor microenvironment. We previously defined the “matrisome” as the ensemble of genes encoding ECM proteins and proteins modulating ECM structure or function. While compositional and biomechanical changes in the ECM regulate cancer progression, no study has investigated the genomic alterations of matrisome genes in cancers and their consequences. Here, mining The Cancer Genome Atlas (TCGA) data, we found that copy number alterations and mutations are frequent in matrisome genes, even more so than in the rest of the genome. We also found that these alterations are predicted to significantly impact gene expression and protein function. Moreover, we identified matrisome genes whose mutational burden is an independent predictor of survival. We propose that studying genomic alterations of matrisome genes will further our understanding of the roles of this compartment in cancer progression and will lead to the development of innovative therapeutic strategies targeting the ECM.

## 1. Introduction

The advent of next-generation sequencing (NGS) techniques and the wealth of “big data” they have generated have revolutionized biomedical research and propelled the discovery of mechanisms underlying diseases [1] leading to the development of novel strategies to diagnose and care for patients. In recent years, The Cancer Genome Atlas (TCGA) has provided researchers with an unmatched set of genomics, epigenomics, transcriptomics, and clinical data [2], enabling disruptive discoveries of driver mutations and oncogenic signaling pathways [3], probing the immune landscape of tumors pathways [4], or correlating genomic alterations to response to anti-cancer therapies [5]. 

While cancer research has mostly focused on the study of tumor–cell-intrinsic processes, the past few decades have seen an increased focus being placed on the study of the tumor microenvironment and on the tumor extracellular matrix (ECM) [6,7,8]. The extracellular matrix (ECM) is the complex and dynamic assembly of hundreds of proteins that regulates cellular metabolism and phenotypes and governs tissue formation and homeostasis [9]. The ECM is a major structural and functional component of the tumor microenvironment [10]. Desmoplasia, or ECM accumulation, is a characteristic feature of tumors, and a higher ECM content is often associated with poorer prognosis in a broad range of cancer types [11]. Moreover, all 10 hallmarks of cancers proposed by Weinberg and Hanahan [12] are under the direct control of chemical or mechanical signals from the ECM [10,13,14]. This recognition of the prominent roles of the ECM in different aspects of cancer progression, including tumor heterogeneity and response to treatment, has been permitted in part by technological advances, including imaging, mechanical probing, and proteomic methods, that have overcome limitations posed by the intrinsic biochemical properties of ECM components [15]. It has also been permitted by the emergence of tools to consistently and comprehensively annotate ECM genes and proteins in big data [16]. To assist with this effort, we previously used a computational approach to predict the “matrisome”, defined as the compendium of genes encoding core ECM proteins, or structural component of the ECM, including collagens, proteoglycans, and glycoproteins, and ECM-associated proteins, including ECM remodeling enzymes, proteins structurally or functionally related to ECM components, as well as secreted factors [17,18]. The matrisome, as a defining framework, has allowed ECM research to enter the -omics era [19]. Used to annotate proteomics data of murine or human tumors, it has revealed that compositional and quantitative alterations of the matrisome contribute to tumor progression [20,21,22,23,24,25]. In addition, proteomics of the ECM of tumor xenografts has also shown that, while stromal cells and in particular cancer-associated fibroblasts are the main contributors to the production of the ECM of tumor microenvironments [26], tumor cells also produce and secrete ECM proteins [17,21,22]. Used to annotate transcriptomic data, it has helped shed light on the ECM contribution to specific cancer types, including high-grade serous ovarian cancer [25,27,28] or acute myeloid leukemia [29], or across cancers [30]. Importantly, in a recent study, we evaluated the level of the expression of matrisome genes in 10,487 patients across 32 tumor types using TCGA data and demonstrated that matrisome gene expression can segregate different tumor types [31]. Last, “omic” technologies have uncovered ECM genes and proteins whose levels are predictive of cancer patient outcome [23,25,32,33,34]. However, while mutations in ECM genes have been linked to a plethora of diseases and syndromes [35], no study has focused on determining the presence and extent of genomic alterations and mutations in ECM genes in cancers, a crucial piece of information to further understanding of the tumor microenvironment (TME) [36].

Here, we sought to profile the genomic and the mutational landscapes of the matrisome using TCGA data. We focused our analysis on a panel of 14 of the most frequent solid cancer types occurring in diverse organs and projected to account for more than 1 million of new cancer cases in the US in 2020 and to be responsible for more than 350,000 cancer deaths in the US in 2020 (Table 1) [37]. For our analysis, we retrieved data on 1014 of the 1027 human matrisome genes for 6740 patients and surveyed the nature and potential consequences of 4433 copy number alterations (CNAs) and 4497 mutations affecting matrisome genes (Table 2). We determined the impact of these genomic alterations (copy number alterations and mutations) on gene expression levels, predicted protein functions, and overall patient survival. Our results demonstrate that matrisome genes are subject to more copy number and mutational alterations than the rest of the genome and that mutations of matrisome genes are statistically more likely to have a functional impact. We further identified common core matrisome and matrisome-associated genes altered across multiple cancer types, and within these genes, we identified sequences encoding protein domains that accumulate more frequent mutations, which hints at the potential functional consequences these mutations could have on the multi-faceted roles of the ECM in cancer initiation and progression. Last, we report the identification of matrisome genes whose mutational burden correlates with overall survival, demonstrating the potential prognostic value of analyzing genomic features of the matrisome to predict cancer patient outcome. 

## 2. Results and Discussion

### 2.1. Copy Number Alterations in Matrisome Genes Are More Frequent than in the Rest of the Genome

We first sought to measure the extent and impact of copy number alterations (CNAs) on matrisome genes. To address this, we evaluated the frequency with which matrisome genes were subject to CNAs and showed that, overall, they tended to be more frequently and/or more extensively altered than the rest of the genome (Graphical Abstract and Figure 1). 

We further determined the type of CNA that affected matrisome genes and stratified CNAs as low- or high-level copy number amplifications (Appendix A) and homozygous or single-copy deletions (Appendix A). To simplify interpretation, we binned data into quartiles (Q0–Q4) based on the percentage of samples showing CNAs in a given tumor, Q0 indicating that no CNAs were detected, Q1 the first quartile where CNAs are found in 0–25% of the samples, Q2 the second where CNAs are found in 26–50% of the samples, etc. Results show that, in general, matrisome CNAs in a tumor follow the same quantitative trends as non-matrisome ones, though at times they are more abundant overall (e.g., Figure 1 BRCA, LUAD, and PAAD) and more quantitatively affected (e.g., gain in Q2 in Figure 1 in LUSC). On the other hand, in some tumors, matrisome CNAs quantitatively decreased (e.g., Figure 1, ESCA, OV, and UCEC). These data suggest a trend towards a more dynamic copy number tolerance than the rest of the genome whose consequences have not been, until now, evaluated.

Further breakdown of the data per matrisome gene category (Appendix A) shows the same general outlook: matrisome genes seem particularly tolerant of copy number alterations, with minor tumor-specific differences in the amount and type of CNAs within the matrisome categories. For example, we observed that the core matrisome (collagens, glycoproteins, and proteoglycans) tends to accumulate more CNAs than matrisome-associated genes in breast and lung neoplasms, while tending to accumulate less in colorectal, ovarian, and uterine tumors, possibly hinting at different contexts, and perhaps selective pressure on the matrisome components, between these tumor types.

### 2.2. Consequences of CNAs on Matrisome Gene Expression Levels

We next sought to determine the potential consequences of CNAs on matrisome gene expression levels. While the majority of alterations were predicted to have no impact on expression levels (Figure 2A), we identified, for each cancer type, subsets of core matrisome and matrisome-associated genes that either have a significantly positive impact on gene expression, i.e., genes showing a 0.5-fold (or 50%) higher expression in patients with CNAs versus patients with no CNAs (Figure 2B,C), or have a significantly negative impact on gene expression, i.e., genes showing a 0.5-fold (or 50%) lower expression in patients with CNAs versus patients with no CNAs (Figure 2B,D). Among these, the majority of genes having a significant impact on gene expression were matrisome-associated genes (Figure 2C,D), suggesting that functional and signaling elements within the matrisome (ECM remodeling enzymes, cytokines and chemokines, growth factors etc.) are more frequently targeted by CNA-dependent expression regulation than structural genes (collagens, glycoproteins, and proteoglycans), which is along the same line of the pan-cancer observations by Shao et al. [38]. Additionally, these findings can also be explained by the high number of paralogs of core matrisome genes that might act as a buffer to preserve the functionality of this compartment.

### 2.3. Matrisome Genes Are Significantly More Susceptible to Be Mutated

Matrisome genes, and in particular core matrisome genes encoding structural components of the ECM such as collagens, are significantly longer than other genes (Figure 3A) and thus call for more mutations than the rest of the genome. This observation prompted us to compute the number of mutations normalized by gene length (see Section 3), which revealed that matrisome genes accumulate significantly more mutations per gene length and across the overall number of genes involved than the rest of the genome (Graphical Abstract, Figure 3B, Appendix A, and Appendix A). This suggests that either or both a lower selective pressure on these mutations by, for example, immune cells and/or a higher fitness as local mutators that act as a buffer to the preservation of the global genomic information might act on matrisome sequences at the genomic level [38,39,40]. In line with this, we found a higher overall mutational burden in the matrisome compartment in comparison to the rest of the genome (Appendix A), but a lower recurrence of mutations, with most mutations in matrisome genes found in only one patient (Appendix A). Of note, for three cancer types, cutaneous melanoma, stomach adenocarcinoma, and endometrial carcinoma, we found a subset of mutations in matrisome genes found in more than five patients (Appendix A). In light of our findings on CNAs, we can speculate that the selective pressure on these mutations might be counteracted and dispersed by the high number of matrisome gene paralogs, which might further point to a role for matrisome gene mutations as local mutators or interactors rather than cancer drivers. 

While most of the mutations identified were specific to only one patient or to a few patients within a single tumor type (Appendix A), we could nonetheless identify potential “hotspot” mutations, defined as occurring in at least five patients per tumor type, and in at least two different tumor types, for six genes (Appendix A). Among these are *PXDN* and *FBN2*, encoding the core ECM glycoproteins peroxidasin and fibrillin 2, whose roles in different cancers have been already discussed [41,42], though no evidence of a mutational impact of these proteins has been presented.

We further interrogated the molecular nature of the mutations and found that for all matrisome gene categories and for most cancer types, these were in majority (>~70%) transitions, i.e., the interchange of a purine for another (A/G) or of a pyrimidine for another (C/T) (Appendix A). Two noticeable deviations are lung adenocarcinomas and skin cutaneous melanomas. Interestingly, for the former, the frequency of transversions, the replacement of a purine by a pyrimidine and a hallmark of the carcinogenic effects of smoking on genes [43], and the frequency of transitions were similar, and this was consistent across all matrisome gene categories (Appendix A). For melanoma, the carcinogenic effects of ultraviolet-A and -B wavelengths suffice to explain the increased amount of transitions, and the sum of these observations put local genomic variance in the matrisome as a factor that probably comes later in carcinogenic evolution than the primary effects of driver events. 

When looking at the type of mutations affecting matrisome genes, we found that the majority of mutations across all cancer types were missense mutations (~50%), followed by silent mutations (~25%) (Figure 3C). We also observed a high percentage of frame shift deletions in breast cancer (BRCA), while cervical squamous cell carcinomas and endocervical adenocarcinomas (CESC) and esophageal carcinomas (ESCA) accumulated a large number of mutations in the 3′ UTR of matrisome genes. In addition, when looking specifically at the frequency of mutation types per matrisome gene categories and cancer types, we observed that matrisome genes and particularly secreted factors presented frequent mutations affecting splicing sites in cervical squamous cell carcinomas and endocervical adenocarcinomas (CESC), esophageal carcinomas (ESCA), and uterine carcinosarcoma (UCS) (Appendix A). This is of particular relevance, since there exist multiple examples of alternative splicing of ECM genes (e.g., fibronectin, tenascin) or growth factors resulting in the production of isoforms only reported to be expressed in pathological conditions such as wound healing and cancers [44,45,46], and these splice variants have been proposed to serve as biomarkers or anchors to selectively target drugs or biological agents to tumors [47,48,49].

### 2.4. Mutated Protein Domains and Potential Consequences on ECM Protein Functions

ECM proteins present a characteristic domain-based organization that supports their scaffolding properties via ECM protein–protein interactions and their signaling properties via ECM/growth factor interactions and ECM/ECM-receptor interactions [18,50,51]. These protein domains initially served as the basis for the in-silico prediction of the matrisome component via sequence analysis [17]. We thus sought to determine whether mutations in matrisome genes occurred preferentially in certain protein domains and/or preferential sites and whether we could infer the possible impact of such mutations on protein folding, protein complex assembly, or signaling functions. We focused our survey on the top 20 most mutated domains in each of the 14 cancer types studies and identified 46 unique protein domains with the highest mutation frequency (Figure 4 and Appendix A). Of these, 19 are core-matrisome-defining protein domains and 15 are matrisome-associated-protein defining domains [17]. While some of these domains are not exclusively found in ECM proteins, others, such as the laminin G and laminin N-terminal domains, the zona pellucida domain, or the NIDO domain, are specific to matrisome components. 

### 2.5. Functional Consequences of Matrisome Gene Mutations 

Using the “Polymorphism Phenotyping v2” (PolyPhen-2) algorithm predictions as previously reported [52], we next evaluated the impact of mutations at the protein level. As compared to mutations affecting non-matrisome genes, we found that mutations of matrisome genes are statistically slightly more likely to have a functional impact (Figure 5A). Further investigations into predicted effects for the different matrisome categories show major differences, with ECM proteins (collagens, proteoglycans, and ECM-affiliated molecules) having a proportionally much greater burden of mutations with unclear/unknown effects as compared to, for example, ECM-associated proteins such as metalloproteinases or growth factors (Figure 5B). 

While further investigations will be required to explain this observation, we can hypothesize that the highly modular structure of core matrisome genes and proteins, in particular collagens, can potentially absorb more mutations without suffering a functional damage with respect to genes with a much simpler organization such as ECM regulators and secreted factors. Moreover, how these mutations affect post-translational modifications and three-dimensional protein conformation remains to be addressed. This general trend holds true when subsetting results by matrisome gene category and cancer type (Appendix A), suggesting no specialization in the functional type of mutations occurring within the matrisome of different tumor types.

### 2.6. Identification of the Top 10 Most Mutated Matrisome Genes across 14 Cancer Types

Our analysis reveals that several matrisome genes are frequently mutated across tumors, though with different specific mutations (Figure 6A and Appendix A). Notably, the ten most mutated matrisome genes per tumor type overlap regardless of tissue- or cell-of origin-patterns (Figure 6A and Appendix A). This is, for example, the case of mucin 16 (*MUC16*) or filaggrin (*FLG*), which are mutated in all 14 tumors analyzed, or of hemicentin 1 (*HMCN1*), mucin 5 B (*MUC5B*) or reelin (*RELN*) mutated in 12/14 tumors. These genes, however, are also the largest of all matrisome genes and it is thus unsurprising to find them topping the Pan-Cancer matrisome mutational burden chart. Interestingly, we observed again a differential distribution in the accumulation of mutations between core matrisome and matrisome-associated genes, with the latter (which includes the mucins) being more frequently represented at top position across all cancers considered.

Among the core matrisome, the most frequently mutated gene across cancer types is *HMCN1*, which encodes the glycoprotein hemicentin-1, also known as fibulin-6 (*FBN6*) and a member of the fibulin protein family and component of basement membranes [53] (Figure 6B). While the importance of *HMCN1* in cancer progression remains unclear, our data suggest a wider contribution to oncological processes than previously reported [54,55]. Of note, our survey did not find any correlation between the mutational burden in *HMCN* and cancer patient survival.

Similarly, especially considering the impact on patient survival (see Figure 7), our data suggest that mutations within the mucin genes, especially *MUC16* and *MUC5B*, are worth further assessment for their potential prognostic use, again expanding on observations from previous reports [56]. 

### 2.7. Consequences of Matrisome Gene Mutations on Patient Survival

Finally, we evaluated the consequences of mutational burden in matrisome genes (at the whole gene level as well as at the domain level) on patient survival, focusing on genes with a concordant effect per se (univariate analysis) and after correcting for age, sex, and ethnicity in multivariate analyses. Figure 7 depicts the prognostic value of two core matrisome genes, *COL6A1* and *LAMB3*, and of two matrisome-associated genes, *MUC5B* and *MUC16*, whose mutational burden significantly correlated either negatively (Table 3A and Table 4A) or positively (Table 3B and Table 4B) with overall survival in at least two cancer types: colorectal cancer and melanoma for *COL6A1*, lung adenocarcinoma and stomach adenocarcinoma for *LAMB3*, melanoma and uterine corpus endometrial carcinoma for *MUC16*, and lung adenocarcinoma and uterine corpus endometrial carcinoma for *MUC5B*. More globally, our results show that, independent of the matrisome category to which these genes belong (core matrisome, Table 3, Figure 7, and Appendix A; or matrisome-associated, Table 4, Figure 7, and Appendix A), the prognostic value of their mutational burden depends on the gene itself, hinting at the functional consequences of mutations on the functions of the respective protein. The same holds true for matrisome protein domains (Appendix A), though, from both the gene-centric and the domain-centric analyses, we observed that mutations in the tumor matrisome are much more likely to associate with increased overall survival (overall survival, approximately 61% of genes and 81% of domains reported in Appendix A), supporting the idea that mutations in the tumor matrisome disrupt the organization of the tumor microenvironment and disadvantage neoplastic cells taking away structural cues they require for extensive growth, spreading, and metastasis. This observation may provide another explanation for the low recurrence of matrisome mutations found across tumors, though the lack of time coordinates within the TCGA data and their bulk rather than single cell structure prevented us from testing this further (see Section 4).

### 2.8. Cross-Validation Using Independent Cancer Patient Cohorts

While cross-cohort comparisons can be hindered by the composition of the cohorts (mixed population background, age, etc.) and the differing end-points used and tend to mask rarer mutations (such as the ones reported here), we further sought to validate our observations in other, comparably large cohorts of cancer patients. We focused on the four genes, *COL6A1*, *LAMB3*, *MUC5B*, and *MUC16*, for which we showed that mutational burden had a prognostic value for patient survival (Figure 7) and interrogated a large collection of samples from patients and cells from 178 cohorts available via the cBioPortal (see Section 3). We observed a wide variation in the number of cases harboring CNAs or mutations in these genes (Appendix A). We also observed an overall low occurrence and recurrence of CNAs and mutations for each of these genes (see the peak of the density plots around the 0 value in Appendix A) and a higher number of studies with cases affected by CNAs or mutations for *MUC5B* and *MUC16*, than for *COL6A1* and *LAMB3* (Appendix A). Importantly, both observations are in line with our findings on matrisome mutational frequencies in TCGA. 

We further sought to cross-validate the prognostic value of these genes in a combined cohort of adult patients from TCGA and Genotype-Tissue expression project (GTEx) cohorts and pediatric patients from the Therapeutically Applicable Research to Generate Effective Treatments (TARGET) cohort and KidsFirst initiatives (Appendix A). Interestingly, we observed significant associations with survival for both *MUC5B* and *MUC16* and a borderline association for *COL6A1,* which is similar to what we observed in the pan-cancer TCGA cohort (Figure 7), with *MUC5B* mutations associating with better survival and *COL6A1* and *MUC1*6 associating with poorer survival (Appendix A).

## 3. Methods

### 3.1. Source Data

All data except the matrisome gene list and Pan-Cancer tumor purity values were sourced from the harmonized TCGA Pan-Cancer Atlas resource and downloaded from the University of California, Santa Cruz UCSC Xena Browser hub (http://xenabrowser.net/). Online analyses for cross-validation purposes were performed through cBioPortal (https://www.cbioportal.org/) and the Xena Browser. The following files were downloaded for further analysis.

### 3.2. Matrisome Gene List

The human matrisome data were downloaded from the Matrisome Project data browser (http://matrisome.org/) [19]. In order to assist with the identification and classification of genes encoding proteins found within the ECM, we previously defined the matrisome as the collection of genes encoding structural elements of the ECM (“core matrisome”) and genes encoding proteins either structurally or functionally associated with the ECM (“matrisome-associated”) [17,18,57]. We further divided these divisions of the matrisome into categories, the core matrisome being composed of collagens, proteoglycans, and other ECM glycoproteins, while the matrisome-associated is composed of proteins structurally of functionally affiliated with ECM proteins, ECM-remodeling enzymes and their regulators (“ECM regulators”), and secreted factors [17,18,57]. 

### 3.3. Gene Expression Data 

Sample-level normalized, log_2_ (norm_value +1)-transformed gene expression data, Xena identifier: EB++AdjustPANCAN_IlluminaHiSeq_RNASeqV2.geneExp.xena were downloaded for this study. The same data are available through Sage Bionetworks’ Synapse Pan-cancer Atlas data browser (http://www.synapse.org/#!Synapse:syn4976369.3). 

### 3.4. Copy Number Alterations (CNAs) 

Gene-level copy number (gistic2_thresholded), Xena identifier: TCGA.PANCAN.sampleMap/Gistic2_CopyNumber_Gistic2_all_thresholded.by_genes. TCGA pan-cancer gene-level copy number alterations (CNA) were estimated using the Genomic Identification of Significant Targets in Cancer 2 (GISTIC2) threshold method, compiled using data from all TCGA cohorts. Copy number was measured experimentally using whole genome microarray at a TCGA genome characterization center. Subsequently, the GISTIC2 method was applied using the TCGA FIREHOSE pipeline to produce gene-level copy number estimates. GISTIC2 further thresholded the estimated values to –2, –1, 0, 1, and 2, representing homozygous deletion, single copy deletion, diploid normal copy, low-level copy number amplification, and high-level copy number amplification, respectively. Genes were mapped onto the human genome coordinates using UCSC cgData HUGO probeMap [58].

Somatic mutations *(SNP* and *INDEL)*: TCGA Unified Ensemble “MC3” mutation calls, Xena identifier: mc3.v0.2.8.PUBLIC.xena [59].

### 3.5. Clinical Data

Curated clinical data, Xena identifier: Survival_SupplementalTable_S1_20171025_xena_sp. These data were derived from the integration of the TCGA Pan-Cancer Clinical Data Resource (TCGA-CDR) [60].

### 3.6. Pan-Cancer Purity Data

TCGA Pan-Cancer tumor purity data (consensus measurement of purity estimations, CPE) were obtained from Aran et al [61].

### 3.7. Cross-Validation Data

The prevalence of mutations in *COL6A1*, *LAMB3*, *MUC5B*, and *MUC16* across 178 studies was evaluated via cBioPortal (http://www.cbioportal.org/) [62,63]. Further assessments on the effect of the mutational burden of these genes on patient survival were conducted in the integrated “TCGA TARGET GTEx KidsFirst” cohort, available via the Xena Browser.

### 3.8. Statistical Analysis

All analyses were performed in The R Project for Statistical Computing (R) and were restricted to the following tumor types: Breast Invasive Carcinoma (BRCA), Cervical Squamous Cell Carcinoma and Endocervical Adenocarcinoma (CESC), Colon Adenocarcinoma (COAD), Esophageal Carcinoma (ESCA), Lung Adenocarcinoma (LUAD), Lung Squamous Cell Carcinoma (LUSC), Ovarian Cancer (OV), Pancreatic Adenocarcinoma (PAAD), Prostate Adenocarcinoma (PRAD), Rectum Adenocarcinoma (READ), Skin Cutaneous Melanoma (SKCM), Stomach Adenocarcinoma (STAD), Uterine Corpus Endometrial Carcinoma (UCEC), and Uterine Carcinosarcoma (UCS). Quantitative categorical differences were tested using a two-sided Chi-square test, while quantitative numerical differences were tested using a two-sided Mann–Whitney U test. 

To calculate the effect of CNAs on transcription, only those CNAs were selected whose expression level for the same gene harboring the CNA in carriers was at least 50% increased or decreased vs. non-carriers. 

Differences in the number of mutations normalized by gene length in the matrisome vs. non matrisome were tested by the Mann–Whitney U test and by further randomization tests. These included (1) 1000 tests against random ~33% of whole non-matrisome human genes, (2) 1000 tests against random non-matrisome human gene sets, each the same size as the number of mutated matrisome genes, and (3) 1000 tests against random non-matrisome human gene sets, each composed of genes longer than the average length of matrisome genes. Gene lengths were pulled from the “Goseq” library in R. For matching mutations and protein domains, we first pulled protein domain coordinates for matrisome genes using the Ensemble database and the R libraries “ensembldb” and “EnsDb.Hsapiens.v86” and then mapped each mutation onto the domains using a “between” SQL query implemented in the R library “sqldf”. 

Hotspots mutations were defined as those occurring at least five times per tumor type in at least two different tumor types. 

Mutation effects on overall survival (at the whole-gene or domain level) were modelled in univariate (Kaplan–Meier) and multivariate (Cox proportional hazard) analyses, the latter including also age at diagnosis, gender, and ethnicity as covariates. Mutations were filtered before analysis to remove entries with fewer than 10 patients in at least one tumor, and analyses were performed for overall survival (OS). Further data can be provided on request to the Authors or by running the relevant code section (see Section 3.9 and Section 3.10). Only genes/domains with a significant concordant effect in both the analyses are reported.

To assess the eventual effect of tumor purity on CNAs and mutations, we retrieved consensus measurement of purity estimations (CPEs, available for 11 of the 14 tumor types studied here) and imputed effects using generalized linear models (GLMs).

In all analyses, a *p* value < 0.05 was chosen as the threshold for reporting significant results.

### 3.9. Data Availability

All starting data are freely available and downloadable from the sources noted above (see Section 3.1). The same data are enclosed in a freely accessible Zenodo repository (10.5281/zenodo.3941354). All results tables can be obtained from the authors upon request. 

### 3.10. Code Availability

All the codes have been prepared into an R notebook and made available through GitHub (http://github.com/Izzilab/pancancer-matrisome-mutations) and Zenodo (10.5281/zenodo.3941348) or as an HTML through RPubs (http://rpubs.com/Izzilab/matrisome-CNAs-and-mutations).

## 4. Conclusions

This first survey of the genomic and mutational landscape of the cancer matrisome has uncovered the interesting, and yet perhaps unexpected, extent and consequences of copy number and mutational alterations of matrisome genes in a panel of 14 solid tumor types. 

Of note, TCGA data were collected from bulk tumor samples and more specifically, mostly from tumor cells, as previously shown [61] and further validated here (Appendix A). We can thus confidently map our findings to tumor cells rather than to other cells of the tumor microenvironment. In this respect, we observed that the presence of eventual impurities (i.e., the presence of other cell types of the TME) is a marginal confounder for CNAs and a negligible one for mutations. Further acknowledging that the number of CNAs and mutations in the cancer genome and the sample composition in terms of tumor/TME fractions are not linearly nor directly associated [61,64], the estimates we report for their interactions probably exceed their true extent.

While we have previously shown that tumor cells do secrete ECM proteins, cancer-associated fibroblasts are the main ECM producers and remodelers. In addition, there is now an increased recognition of the impact of cellular and microenvironmental heterogeneity on tumor progression, metastasis formation, and response to treatment. Future studies should thus focus on elucidating the presence and roles of matrisome CNAs and mutations in the different cell populations found in the tumor microenvironment, the timeline of their occurrence, and, importantly, the loco-regional distribution of mutated ECM proteins within the tumor microenvironment. 

Future studies are also necessary to decipher the functional consequences of the mutations identified here. One possibility is that mutations in matrisome genes, and more specifically located in sequences encoding protein domains, can affect protein/protein (e.g., ECM protein/ECM protein, ECM/growth factor, ECM/enzyme, ECM protein/ECM receptor) interactions and subsequently alter biochemical and mechanical signaling, leading to dysregulation of cellular phenotypes and eventually to cancer progression. Additionally, with recent reports highlighting the impact of the ECM on immune cells within the tumor microenvironment [30], mutations in matrisome genes could also result in the generation of neo-antigens and thus rewire the immune response.

Last, our analysis also had the power to identify matrisome genes whose mutational burden was an independent predictor of overall survival. It would thus be interesting to expand our survey to the study of specific genes that could predict disease-specific or metastasis-free survival and compute whether mutational burden in certain matrisome genes correlates with the variation of the progression-free interval. 

We believe that our results are a starting point to the more extensive mapping of clinically relevant matrisome gene alterations and can be used to prioritize further investigations that may lead to significant translational applications to improve cancer patient care.

## Figures and Tables

**Figure 1 cancers-12-02046-f001:**
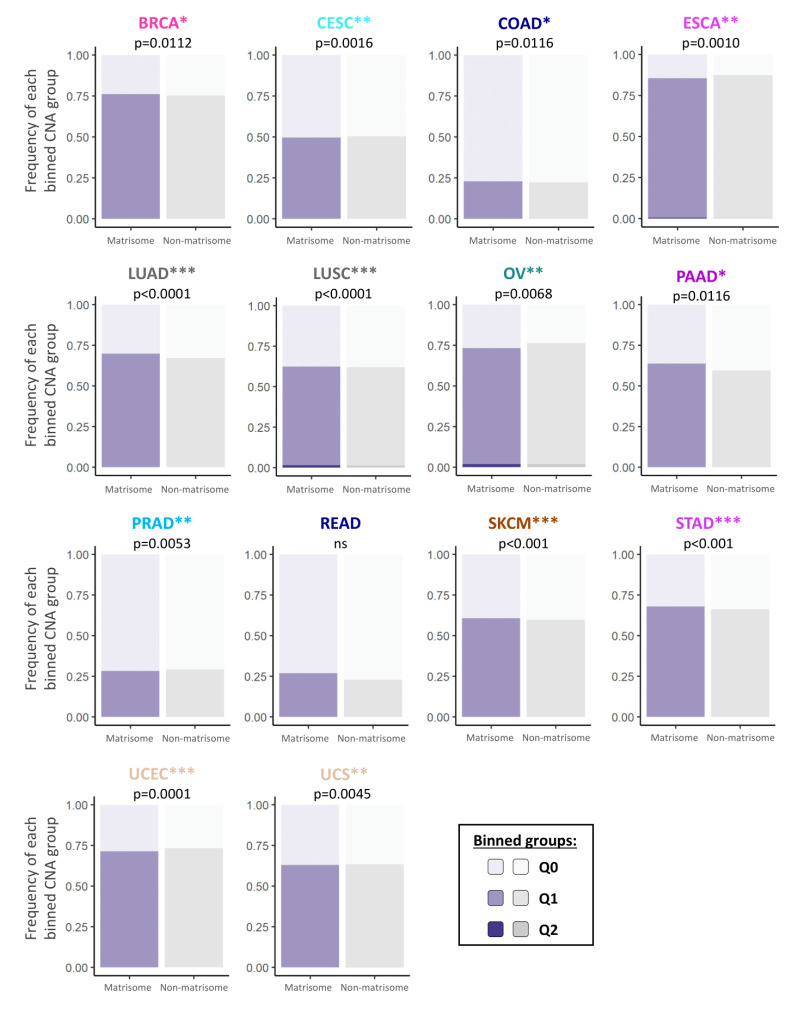
Frequency of copy number alterations of matrisome genes across 14 different cancer types. Bar charts represent the frequency of CNAs in matrisome genes (purple bars) and non-matrisome genes (rest of the genome, grey bars) across 14 different cancer types. Chi-square test *p*-values, (calculated on the frequency per binned categories) are indicated for each cancer type (* *p* < 0.05; ** *p* < 0.01; *** *p* < 0.001). Binned groups are represented by different shades of purple and grey to represent genes in which CNAs are found in x% of the samples: Q0 = 0% (lighter shade), 0% < Q1 ≤ 25%, 25% < Q2 ≤ 50% (darker shade). See also Appendix A.

**Figure 2 cancers-12-02046-f002:**
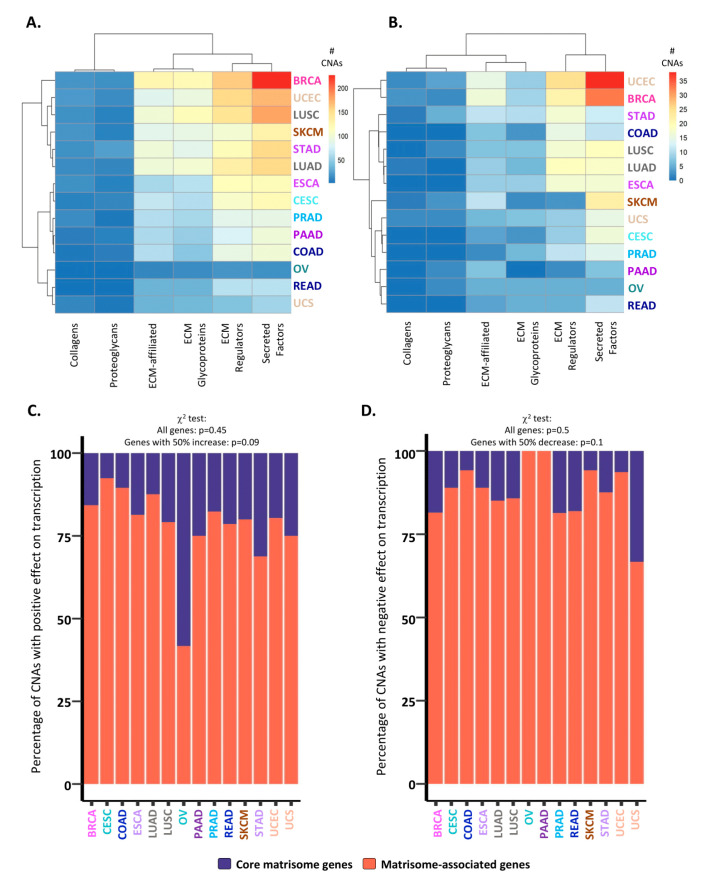
Consequences of CNAs on matrisome gene expression levels. (**A**) Number of total CNAs identified in matrisome genes per cancer type and matrisome category. (**B**) Number of CNAs with significant effects (i.e., resulting in at least a 50% increase or decrease in expression level vs. patients with no CNAs) on matrisome gene transcription per cancer type and matrisome category. In (**A**) and (**B**), bars represent the number of genes whose expression is affected by CNAs per tumor, from lowest (blue) to highest (red) values. (**C**,**D**) Bar charts represent the percentage of CNAs with a significant (>50% higher or lower gene expression than in patients with no CNA) positive (**C**) or negative (**D**) impact on core matrisome (purple) or matrisome-associated (coral) gene transcription.

**Figure 3 cancers-12-02046-f003:**
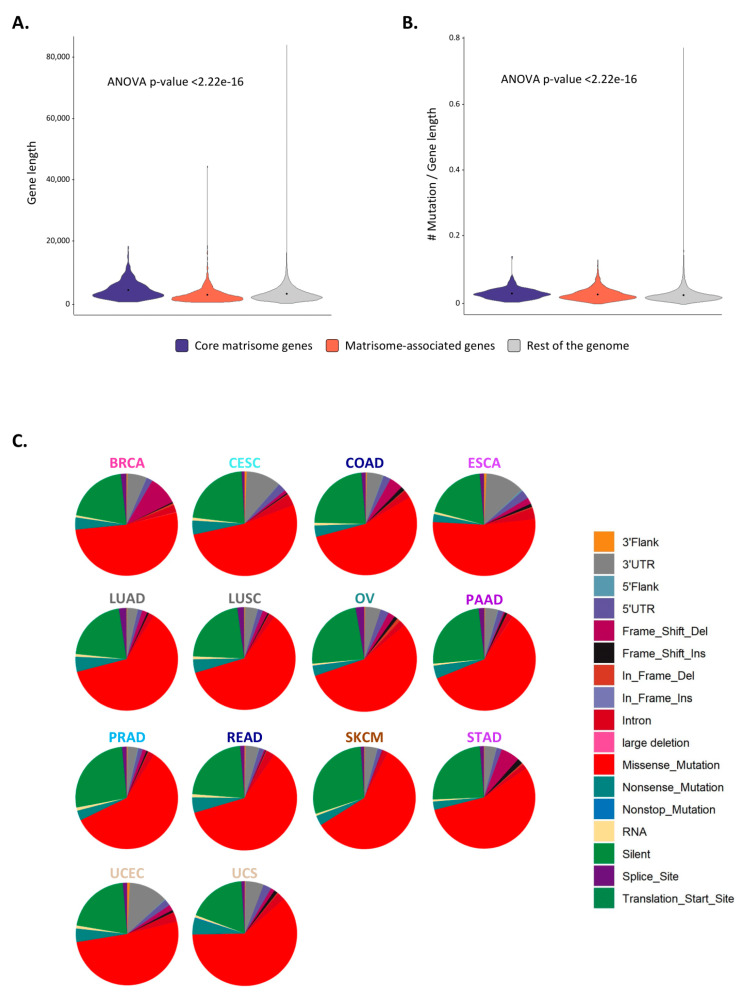
Mutations in matrisome genes (**A**) Violin plot represents the density of genes of given lengths for core matrisome (purple), matrisome-associated (coral), and non-matrisome (grey) genes. Dots indicate the mean of the distribution. (**B**) Violin plot represents the density of genes of given number of mutations *per* gene length ratios for core matrisome (purple), matrisome-associated (coral), and non-matrisome (grey) genes. Dots indicate the mean of the distribution. (**C**) Pie charts represent the type of mutations in matrisome genes across cancer types. See also Appendix A.

**Figure 4 cancers-12-02046-f004:**
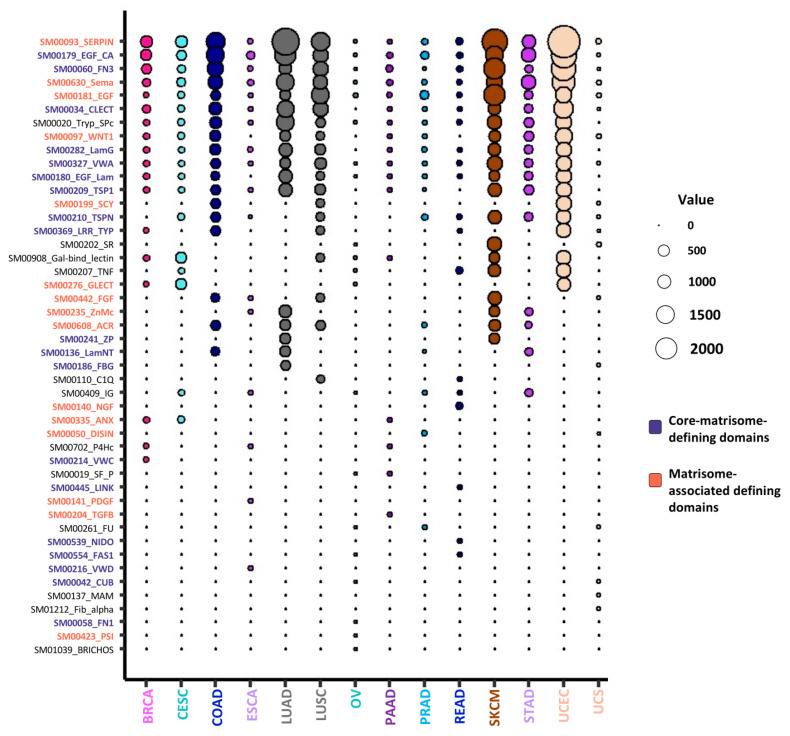
Top 20 most frequently mutated domains in extracellular matrix (ECM) proteins. Bubble plot represents the top 20 most frequently mutated domains in ECM proteins across all 14 cancer types analyzed. The color code indicates domains originally used to predict core-matrisome-proteins (purple) and matrisome-associated (coral) proteins (see [17] for details on matrisome-defining domains). The diameter of the bubbles is proportional to the mutation frequency of each domain in each cancer type. See also Appendix A.

**Figure 5 cancers-12-02046-f005:**
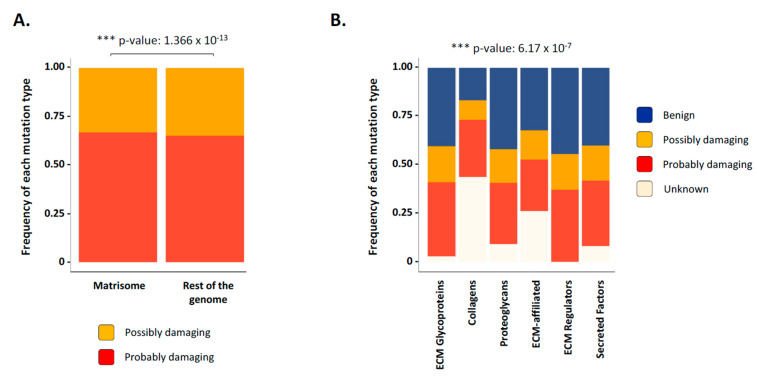
Prediction of mutational effects of matrisome genes on function. (**A**) Bar chart presents the frequency of possibly damaging (yellow) or probably damaging (red) mutations in matrisome genes and non-matrisome genes across all cancer types studied. (**B**) Bar chart presents the frequency of the predicted mutational effect of matrisome genes on function: benign (blue), possible damaging (yellow), probably damaging (red), and unknown (light pink) across all cancer types studied. See also Appendix A.

**Figure 6 cancers-12-02046-f006:**
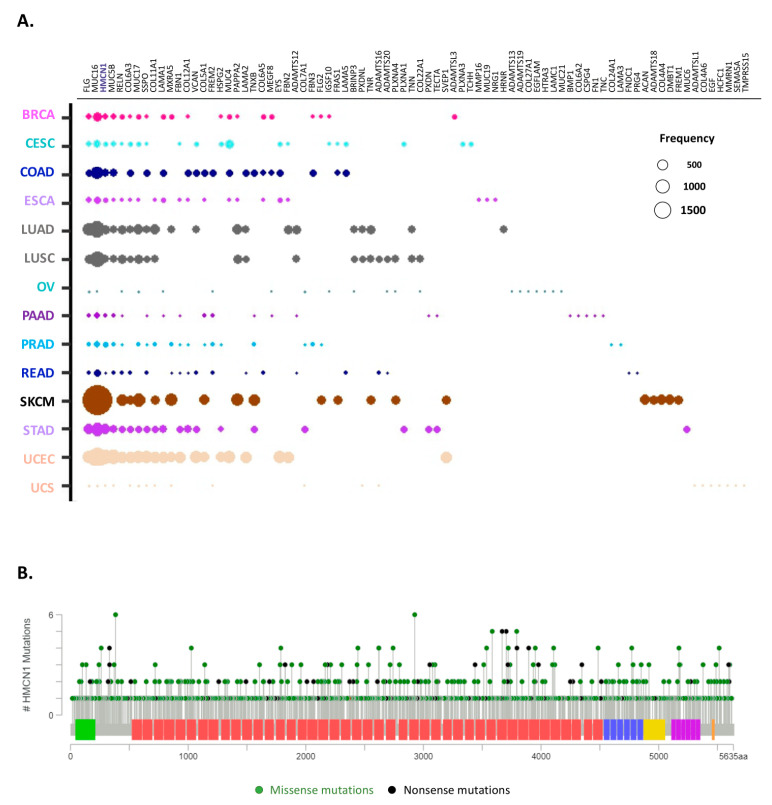
Top 10 most mutated matrisome genes. (**A**) Bubble plot represents the top 10 most frequently mutated genes in ECM proteins across all 14 cancer types analyzed. The diameter of the bubbles is proportional to the mutation frequency of each gene in each cancer type while the relative position along the x axis reflects the number of tumor types in which the gene is mutated (the leftmost genes being mutated in all tumor types analyzed). See also Appendix A. (**B**) Lollipop charts show the mutational landscape for the most frequently mutated core matrisome gene, *HMCN*, encoding hemicentin. Missense mutations (green circles) and truncation mutations, including nonsense mutations, nonstop mutations, frameshift deletions, frameshift insertions or splice sites (black circles), are shown. Data and lollipop graphs were obtained from cBioPortal.

**Figure 7 cancers-12-02046-f007:**
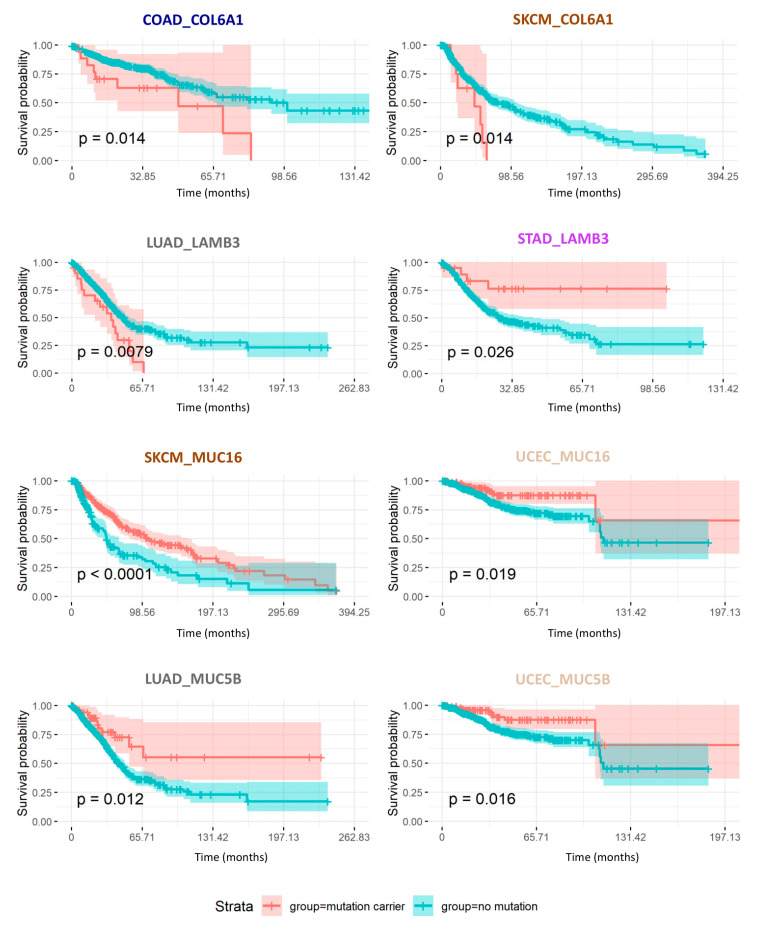
Mutation burden in core matrisome genes impact cancer patient overall survival. Kaplan–Meier curves represent the overall survival probability over time (in months) of patients carrying (coral trace) or not (teal trace) mutations in the specified core matrisome genes *COL6A1* and *LAMB3*, or matrisome-associated genes MUC5B or MUC16. P-values indicated correspond to the ones calculated in the univariate analysis. See Table 3 and Table 4.

**Table 1 cancers-12-02046-t001:** List of cancer types included in the meta-analysis.

Abbreviation	Cancer Type	Estimated New Cases in 2020 in the US	Estimated Deaths in 2020 in the US	5-Year Survival (2009–2015)
**BRCA**	Breast Carcinoma	279,100	42,690	91%
**CESC**	Cervical Squamous Cell Carcinoma and Endocervical Adenocarcinoma	13,800	4290	69%
**COAD/READ**	Colon Adenocarcinoma/Rectum Adenocarcinoma	147,950	53,200	66%
**ESCA**	Esophageal Carcinoma	18,440	16,700	21%
**LUSC/LUAD**	Lung Squamous Cell Carcinoma/Lung Adenocarcinoma	228,820	135,720	21%
**OV**	Ovarian Serous Cystadenocarcinoma	21,750	13,940	48%
**PAAD**	Pancreatic Adenocarcinoma	57,600	47,050	10%
**PRAD**	Prostate Adenocarcinoma	191,930	33,330	99%
**SKC**M	Skin Cutaneous Melanoma	100,350	6850	94%
**STAD**	Stomach Adenocarcinoma	27,600	11,010	32%
**UCS/UCEC**	Uterine Carcinosarcoma/Uterine Corpus Endometrial Carcinoma	65,620	12,590	83%
	**Total**	**1,152,960**	**377,370**	

Note: Cancer types are color-coded using the color of their respective awareness ribbon.

**Table 2 cancers-12-02046-t002:** Number of patients included in the meta-analysis and number of patients for which copy number alterations (CNAs) or mutations in matrisome genes were found.

Abbreviation	Cancer Type	# Of Patients in TCGA	# Of Patients with Matrisome CNAs	# Of Patients with Matrisome Mutations
**BRCA**	Breast Carcinoma	1236	773 (63%)	749 (61%)
**CESC**	Cervical Squamous Cell Carcinoma and Endocervical Adenocarcinoma	312	276 (88%)	278 (89%)
**COAD/READ**	Colon Adenocarcinoma/Rectum Adenocarcinoma	545/183	270 (50%)/87 (48%)	288 (53%)/87 (48%)
**ESCA**	Esophageal Carcinoma	204	181 (89%)	183 (90%)
**LUAD/LUSC**	Lung Adenocarcinoma/Lung Squamous Cell Carcinoma	641/623	504 (79%)/473 (76%)	506 (80%)/475 (76%)
**OV**	Ovarian Serous Cystadenocarcinoma	604	59 (10%)	58 (10%)
**PAAD**	Pancreatic Adenocarcinoma	196	156 (80%)	155 (79%)
**PRAD**	Prostate Adenocarcinoma	566	447 (79%)	437 (77%)
**SKCM**	Skin Cutaneous Melanoma	479	359 (75%)	356 (74%)
**STAD**	Stomach Adenocarcinoma	511	424 (83%)	429 (84%)
**UCEC/UCS**	Uterine Corpus Endometrial Carcinoma/Uterine Carcinosarcoma	583/57	368 (63%)/56 (98%)	440 (75%)/56 (98%)
	**Total**	**6740**	**4433 (66%)**	**4497 (67%)**

Note: Cancer types are color-coded using the color of their respective awareness ribbon.

**Table 3 cancers-12-02046-t003:** Mutated core matrisome genes impacting patient survival.

A. Mutated core matrisome genes with a negative impact on overall survival based on univariate and multivariate analyses						
**Tumor**	**Gene**	**Matrisome Category**	**OS Difference**	***p*** **-Value, Univariate**	***p*** **-Value, Multivariate**	**# Cases with Mutations**
COAD	*OTOL1*	ECM Glycoproteins	−1.811	0.021	0.010	10
COAD	*MATN2*	ECM Glycoproteins	−1.675	0.040	0.048	11
COAD	*NELL2*	ECM Glycoproteins	−1.271	0.018	0.012	16
COAD	*LTBP4*	ECM Glycoproteins	−1.040	0.010	0.011	23
LUAD	*MMRN2*	ECM Glycoproteins	−1.910	0.000	0.000	11
LUAD	*LAMC2*	ECM Glycoproteins	−1.841	0.017	0.016	11
LUAD	*COL22A1*	Collagens	−1.595	0.012	0.009	63
LUAD	*LAMB3*	ECM Glycoproteins	−1.106	0.008	0.011	22
LUSC	*CILP2*	ECM Glycoproteins	−2.269	0.036	0.026	14
LUSC	*COL2A1*	Collagens	−1.099	0.010	0.026	22
PRAD	*MXRA5*	ECM Glycoproteins	−1.146	0.007	0.038	10
SKCM	*COL6A1*	Collagens	−1.478	0.014	0.028	10
B. Mutated core matrisome genes with a positive impact on overall survival based on univariate and multivariate analyses						
**Tumor**	**Gene**	**Matrisome Category**	**OS Difference**	***p*** **-Value, Univariate**	***p*** **-Value, Multivariate**	**# Cases with Mutations**
COAD	*COL6A1*	Collagens	1.026	0.014	0.036	17
LUAD	*TNC*	ECM Glycoproteins	1.382	0.014	0.040	18
LUAD	*MMRN1*	ECM Glycoproteins	1.414	0.017	0.029	43
LUSC	*COL25A1*	Collagens	1.860	0.015	0.038	16
SKCM	*ACAN*	Proteoglycans	1.405	0.025	0.006	64
SKCM	*COL4A6*	Collagens	1.579	0.010	0.007	48
SKCM	*HSPG2*	Proteoglycans	1.650	0.029	0.021	38
SKCM	*COL4A3*	Collagens	1.755	0.005	0.004	49
STAD	*COL15A1*	Collagens	1.201	0.015	0.012	31
STAD	*VWF*	ECM Glycoproteins	1.215	0.015	0.012	32
STAD	*TECTA*	ECM Glycoproteins	1.242	0.009	0.014	41
STAD	*NELL2*	ECM Glycoproteins	1.398	0.004	0.023	19
STAD	*COL4A1*	Collagens	1.495	0.049	0.026	31
STAD	*LAMB3*	ECM Glycoproteins	1.568	0.026	0.030	22
STAD	*COL5A2*	Collagens	1.720	0.012	0.009	22
UCEC	*FBN2*	ECM Glycoproteins	1.059	0.005	0.036	79
UCEC	*RELN*	ECM Glycoproteins	1.163	0.010	0.044	61
UCEC	*FRAS1*	ECM Glycoproteins	1.259	0.015	0.037	66

**Table 4 cancers-12-02046-t004:** Mutated genes encoding extracellular matrix regulators or affiliated proteins impacting patient survival.

A. Mutated genes encoding ECM regulators or ECM-affiliated proteins with a negative impact on overall survival based on univariate and multivariate analyses						
**Tumor**	**Gene**	**Matrisome Category**	**OS Difference**	***p*** **-Value, Univariate**	***p*** **-Value, Multivariate**	**# Cases with Mutations**
BRCA	*SULF2*	ECM Regulators	−1.443	0.037	0.047	10
COAD	*ADAMTS15*	ECM Regulators	−1.238	0.002	0.003	12
COAD	*GPC6*	ECM-affiliated	−1.100	0.031	0.043	15
COAD	*GPC5*	ECM-affiliated	−1.009	0.039	0.046	11
LUAD	*FCN2*	ECM-affiliated	−1.473	0.047	0.036	13
LUAD	*ADAM19*	ECM Regulators	−1.340	0.012	0.012	31
LUAD	*PLXNA4*	ECM-affiliated	−1.065	0.041	0.040	52
LUAD	*MMP16*	ECM Regulators	−1.037	0.015	0.029	54
LUSC	*PLG*	ECM Regulators	−1.770	0.042	0.008	14
LUSC	*ITIH6*	ECM Regulators	−1.565	0.028	0.016	25
SKCM	*PZP*	ECM Regulators	−1.565	0.022	0.031	33
SKCM	*CLEC6A*	ECM-affiliated	−1.460	0.028	0.013	18
SKCM	*SEMA5A*	ECM-affiliated	−1.252	0.022	0.035	26
B. Mutated genes encoding ECM regulators or ECM-affiliated proteins with a positive impact on overall survival based on univariate and multivariate analyses						
**Tumor**	**Gene**	**Matrisome Category**	**OS Difference**	***p*** **-Value, Univariate**	***p*** **-Value, Multivariate**	**# Cases with Mutations**
BRCA	*PLXNA2*	ECM-affiliated	1.011	0.013	0.030	15
LUAD	*MUC5B*	ECM-affiliated	1.296	0.012	0.014	53
LUAD	*ADAMTS5*	ECM Regulators	1.342	0.015	0.016	32
LUSC	*ADAMTSL1*	ECM Regulators	1.624	0.008	0.025	16
LUSC	*TGM7*	ECM Regulators	2.123	0.009	0.043	10
SKCM	*FREM2*	ECM-affiliated	1.371	0.036	0.004	51
SKCM	*COLEC12*	ECM-affiliated	1.703	0.047	0.031	20
SKCM	*MUC16*	ECM-affiliated	2.100	0.000	0.000	251
STAD	*MUC16*	ECM-affiliated	1.077	0.040	0.027	145
STAD	*ADAMTSL3*	ECM Regulators	1.186	0.025	0.029	25
STAD	*SULF1*	ECM Regulators	1.246	0.025	0.035	30
STAD	*MUC4*	ECM-affiliated	1.301	0.027	0.024	29
STAD	*CSPG4*	ECM-affiliated	1.551	0.026	0.020	26
STAD	*ADAM12*	ECM Regulators	1.913	0.046	0.042	12
STAD	*MMP3*	ECM Regulators	1.916	0.031	0.026	13
STAD	*SERPINB8*	ECM Regulators	2.546	0.014	0.027	10
UCEC	*MUC5B*	ECM-affiliated	1.128	0.019	0.043	102
UCEC	*PLXNB3*	ECM-affiliated	1.154	0.026	0.050	60

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
