# Peer review of "Pan-Cancer Analysis of the Genomic Alterations and Mutations of the Matrisome"

_cancers, 2020, doi:10.3390/cancers12082046_

Round 1

Reviewer 1 Report

General comments

In this manuscript, Izzi and colleagues present a large bioinformatics study investigating the frequency of matrisome CNA/mutations in a panel of 14 cancer types available in TCGA dataset, as well as interrogating their role in gene expression level, and overall survival. The authors have presented the following findings:

  • CNA and gene/domain mutations are slightly more frequent in matrisome genes even after normalization for gene length (as matrisome genes are generally larger). Yet statistically significant for some cancer types, this difference in CNA/mutation frequency is really not very pronounced, although the authors are overestimating these findings a bit. The authors explain this observation by limited functional significance of these mutations which creates little difference, if any, in terms of cancer evolution, and therefore negligible mutation pressure.
  • Matrisome mutations have little recurrence in different cancer patients, thus there are no oncogenes/driver mutations among matrisome genes.
  • The list of top 20 matrisome domain/genes is presented
  • Some of these mutated proteins predict improved/worsened overall survival in cancer patients

The manuscript is overall professionally written, and the group has significant expertise in matrix biology. I would therefore recommend its publication.

Specific comments:

0. I would recommend modifying the Fig. 1 to make it more visually impactful. As is, the Fig. 1 has the following issues:

  • From figure legends and results section, it is unclear what left and right bars indicate, for each chart. Legend at the bottom is counterintuitive. Maybe use some sort of labels as “ECM” and “Other” for every single chart?
  • Q2 (dark gray)or Q3 or Q4 bin do not really exist in the chart, even though they mentioned in figure legend and panel legend bottom right corner. The chart is only represented by Q0 and Q1 bins.
  • Statistical significance – what exactly are you comparing using Chi-square, categories or bins? Specify in figure legends.
  • The main conclusion of Figure 1 is that even though matrisome represents ~5% of human genome, the frequency of CNAs in matrisome genes exceeds those of the rest of the genome. This is quite a strong statement. I would recommend the authors to create some kind of graph to better illustrate this finding (as it is difficult to get from current Fig. 1).
  • The difference between Q2 bins between “ECM” and “Other” groups does not seem to be that dramatic. In some cancer types it is the opposite (OV, ESCA, SESC), i.e.  Other  group has more Q2 CNAs than ECM group. I am interested, if you combine these cancer types into one chart, will there be a difference in CNA frequency?
  • Labeling different cancer types with different colors may create an impression these differences are meaningful. In fact, they are not
  1. The authors have found that in matrisome-associated genes, CNAs have more significant effects on gene transcription. On the opposite, core matrisome CNA had very little effect on gene expression. The authors explain it as follows: “functional and signaling elements within the matrisome (ECM remodeling enzymes, cytokines and chemokines, growth factors etc.) are more frequently targeted by CNA-dependent expression regulation than structural genes (collagens, glycoproteins and proteoglycans)”. I disagree there is preferential “targeting”. The more plausible explanation is that structural ECM genes have much less functionality while having multiple paralogs, than ECM-associated proteins many of which are signaling molecules. It could be other explanations. I encourage the authors to provide more speculations on explaining this observation, as it is potentially remarkably interesting.
  2. In Fig. 3A and B, it is unclear whether dots indicate mean or median (specify in legends). In any case, it is difficult to visually compare these violin plots and see any difference.
  3. Why is Fig. 6A having many more genes than 10 as indicated in the legend? It is difficult to see gene names and I suggest the authors to exclude the last 20-25 genes in the list, since their mutation rate is negligible and does not make a difference. Also, the figure legends for Fig. 6A says “Bubble plot represents the top 10 most frequently mutated domains in ECM proteins across all 14 cancer types analyzed”. In fact, mutated domains were analyzed in figure I think it should be genes, not domains.
  4. Figure 7, what is the unit for time? Please indicate units.
  5. Data presented in Fig.5 is highly speculative. The distribution in Fig.5B is a bit random. I don’t feel this analysis adds much to the plot of the paper. From data presented in Fig.5A, it is hard to believe that “mutations of matrisome genes are statistically more likely to have a functional impact”. It also contradicts authors’ conclusion that genetic variations of cancer matrisome are experiencing little selection pressure and therefore do not recur in different patients of one cancer type. Thus, I’m not sure if this all fits together. Graph color and legend color does not match.

Reviewer 2 Report

Specific comments:
1) Perhaps the presentation of the frequencies of CNAs in matrisome and non-matrisome genes could be clearer if the authors used boxplots, instead of bar charts.
2) Please explain why it is believed that matrisome CNAs tend to be more abundant in BRCA, LUAD, and PAAD tumors.
3) Please include Q3 and Q4 in the legend of Figure 1.
4) Please explain which genes belong to core matrisome and which to matrisome-associated categories.
5) Several sentences are confusing and need to be rephrased (i.e., "genes shown to have a greater than 50% lower expression in ..."). I would recommend language editing.
6) in Fig. 2A-B, what do the blue and red colors in the HCL stand for? Please explain.
7) I couldn't locate the data and codes on Zenodo repository (10.5281/zenodo.3889651).
8) I would recommend expanding the Discussion section.
9) The authors seem to have normalized the number of mutations by gene length but did they also do this for the CNAs, and if so, how?

Reviewer 3 Report

This is a very well scheduled and performed study on the correlation of the matrisome with cancerogenesis and patient outcome.

Recent developments highlight indeed, the potential importance of the ECM gene signature in the pathogenesis of cancer. The present study is a significant step in this direction.

The conclusion/discussion section could be supported by further bibliographic data.

Reviewer 4 Report

The manuscript entitled “Pan-cancer analysis of the genomic alterations and mutations of the matrisome” by Izzi V., Davis M. and Naba A., is an appealing and smartly crafted in-silico study, in which the authors analyzed TCGA’s gene expression profiles covering 14 of the most prevalent cancer types. The authors report interesting and relevant data regarding the presence of copy number alterations and mutations in matrisome genes is a common trait among the cancers evaluated, pinpointing to particular altered genes which suggest having an important impact on the over all survival of patients.

However, and despite their relevant nature of their discoveries, the authors failed in vetting their results, which could support better their findings and potentiate the impact of their research. Therefore a more compressive evaluation of their data, supported by a stronger interpretation and discussion of their findings, are needed to better sustain this study.

As a reviewer, my mayor concerns deal with the following:

  1. There is an urgent need for validation of the main discoveries (e.g. mutation burden of Col6A1, LAMB3, MUC16, MUC5B, etc.) sourcing other data bases.
  2. Authors are urged to verify and corroborate, the protein expression profile of these altered genes in human tissue (e.g. IHC using representative TMAs), correlating their findings with their corresponding clinical data (e.g. OS).
  3. Although the authors acknowledged that TCGA data is mostly generated by cancer cells within tumors, it is important to know the percentage (or ratio) of cancer/stroma content of the tumors sourced from the TCGA. Additionally, their discoveries need to be discussed in more detail within this context.
  4. The authors pointed to interesting findings related to the gene HMCN1 which later were not assessed within the mutation burden impact of matrisome genes on the patient overall survival analysis. Authors are encouraged to discuss the rationale for not assessing this, or else to perform that analysis, including the verification mentioned in points 1 and 2.

Minor points are listed:

  1. In Figure 2, please include details regarding the magnitude scale bar provided (in tones from blue to red) within the figure legend.
  2. In Table 2, please include the corresponding percentage (%) of cases annotated in columns: “# of patients with matrisome CNAs” and “# of patients with matrisome mutations”
  3. Are the data plotted in Fig. 2C and D, Fig. 5B, Supp. Fig. 4, 5 and 6 significant? If so please annotate the significance.
  4. In Figure 7, please instead of expressing the x axis (time) in days, use months.

Round 2

Reviewer 4 Report

I am pleased to read that Authors have provided substantial rationale to support their findings.

They have discussed valid points to exclude certain analysis that I proposed. After reading their replies, I certainly agree with the fact that resources and adequate tools needed to evaluate delicate intricacies of their discoveries (e.g. tissue IHC studies), poise a challenge that not necessarily would render a greater advantage at this point. I agree with them that these details could now be explored and dissected in a future publication, which I would be very excited to read.

Moreover, Authors also have provided additional controls and supportive material as requested, which now make a stronger case for their discoveries. All these improvements are also reflected in their results description, as well as in their conclusions.